# From Cultural Safety to Anti-Racism: Reflections on Addressing Inequities in Palliative Care

Seana Bulle [1], Amit Arya [2,3,4,5] and Naheed Dosani [6,7,8,9,*]

1  Department of Family & Community Medicine, University of Toronto, 500 University Avenue, Toronto, ON M5G 1V7, Canada; seana.bulle@unityhealth.to
2  Palliative Care Physician, Freeman Centre for the Advancement of Palliative Care, North York General Hospital, 4001 Leslie Street Ontario, North York, ON M2K 1E1, Canada; amit.arya@nygh.on.ca
3  Specialist Palliative Care in Long-Term Care Outreach Team, Kensington Health, 25 Brunswick Avenue, Toronto, ON M5S 2L9, Canada
4  Division of Palliative Care, Department of Family & Community Medicine, University of Toronto, 500 University Avenue, Toronto, ON M5G 1V7, Canada
5  Division of Palliative Care, Department of Family Medicine, McMaster University, 100 Main Street West, Hamilton, ON L8P 1H6, Canada
6  Department of Family & Community Medicine, St. Michael's Hospital, Unity Health Toronto, 36 Queen Street East, Toronto, ON M5B 1W6, Canada
7  PEACH (Palliative Education And Care for the Homeless) Program, Inner City Health Associates, 145 Front Street East, Toronto, ON M5A 1E3, Canada
8  Kensington Hospice, Kensington Health, 45 Brunswick Avenue, Toronto, ON M5A 3M3, Canada
9  Canadian Partnership against Cancer, 145 King Street West, Toronto, ON M5H 1J8, Canada
*  Correspondence: naheed.dosani@unityhealth.to

**Abstract:** The purpose of palliative care is to ease the suffering of individuals with a serious and often life-limiting illness throughout the course of their disease by providing holistic care that considers the physical, spiritual, and psychosocial dimensions of health and well-being. Research shows that a palliative approach to care is cost-effective for the healthcare system and results in improved quality of life for patients and their loved ones. However, it is well-documented in the literature that structurally vulnerable populations have greater difficulty accessing equitable and culturally safe palliative care. Several domains are identified as contributing factors to the disparities seen in the literature, including systemic racism, cultural differences around death and suffering, and language barriers. Although Canada has had a national palliative care framework since 2018, ongoing issues of access and equity continue to disproportionately impact certain groups, including racially marginalized, immigrant, and low-income communities. In this commentary, successes and ongoing gaps in providing culturally safe and anti-racist palliative care are explored. In these proposed interventions, we advocate for a palliative approach to care that is grounded in equity, justice, and human rights.

**Keywords:** cultural safety; health equity; healthcare accessibility; anti-racism; palliative care

## 1. Introduction

During the first year of the COVID-19 pandemic, Black people in Ontario, Canada, had the highest rates of mortality from COVID-19 (almost two times higher than White residents), followed by those who identified as South Asian and Chinese [1]. White Ontarians, on the other hand, experienced the lowest rates of hospitalization, intensive-care unit (ICU) admission, and death [2]. The COVID-19 pandemic exposed and magnified many health disparities, including vast differences in palliative care for racially marginalized people living in Canada.

Black, South Asian, East Asian, and other groups that face racism have been shown to have higher rates of aggressive end-of-life care in hospitals and limited interactions with

palliative care services. The aforementioned COVID-19 health disparities are only a few examples of how inequities in life can extend through to increased suffering that people also experience as they approach end-of-life.

This commentary will discuss how social, cultural, and historical factors continue to profoundly shape an individual's illness experience and approach to palliative and end-of-life care [3]. By understanding the social and cultural factors that impact a patient's illness course, providers can learn to recognize their own role in addressing ongoing disparities, increase their advocacy to improve health equity, and commit to engaging in culturally safe and anti-racist practice.

## 2. Cultural Barriers in Palliative Care Contribute to Systemic Racism

Palliative care as a specialty typically embodies beliefs, policies, and practices that encompass Western perspectives of death and dying [4,5]. Due to having primarily white founders, Western approaches to palliative care are more likely to be culturally accessible and culturally safe for the groups it was inherently designed to serve [6]. Moreover, it is important to understand the historical impacts of broader discrimination and colonialism (e.g., sterilization of more than 1200 Indigenous people) that have contributed to ongoing mistrust in the Canadian healthcare system at large (which includes palliative care) for many racially marginalized communities [5,7]. The Canadian literature supports the fact that cultural barriers exist in palliative care for a number of groups, including Chinese, South Asian, Black, immigrant, and Sikh communities [8–10].

For example, descendants of Chinese and South Asian ethnicity, South Asian immigrants, and recent immigrants from minority ethnicities had the highest rates of aggressive end-of-life care and were more likely to die in an intensive care unit [8–10]. Another study showed that recent immigrants were 15% more likely to receive aggressive care at end-of-life and 5% less likely than long-term residents (immigrated prior to 1985) to receive supportive care [11]. Southeast Asians had a 25% lower likelihood of receiving supportive care, while White Western Europeans had a 16% higher likelihood [11]. Studies in the United Kingdom and United States also support this data that minority ethnic groups are often underrepresented in palliative care due to experiences of racism, discrimination, and disparities in accessing appropriate care [12,13].

Patients of different cultural backgrounds often have differing views of pain management, family involvement, communication, and preference for location of death (hospice, home care, or hospital setting) [6,14,15]. In some cultures, pain may be viewed as a part of the dying process and tolerated, resulting in lower uptake of pain medicine [16]. Some cultures discuss pain and end-of-life decisions collectively, while others may leave the choice to the spouse, or children [16]. Many cultures use indirect communication rather than direct communication, such as nonverbal behaviors or tone of voice, which requires an understanding of the underlying background or context [17].

Limited access to or use of translators and lack of representation of healthcare workers also perpetuate barriers. It is not the current standard of care to ensure that interpretation services are used every time a healthcare provider speaks with a non-majority language speaker [5]. In a study on home care recipients admitted to hospital in Ontario, allophone patients (patients speaking a language other than French or English) who received language-concordant care had a lower risk of adverse events and in-hospital death, as well as shorter stays in hospital than allophone patients who received language-discordant care [18].

The lower rates of use of palliative care by nonwhite racial groups are not due to preference differences across groups, but rather, differences in access to care and in how that care is provided [19,20].

For example, studies show that South Asians are less likely to receive specialist palliative care [21,22]. One observational study in Canada found that 70% of South Asians had not heard of palliative care, and only 27% thought that it represented their cultural values [8,23]. A vast majority expressed a desire for more information on palliative care altogether [8,23].

Unsurprisingly, people who are socially disadvantaged are at higher risk of cancer-related mortality and are often diagnosed with more advanced cancers than other people, and earlier, too [24,25]. In Canada, Black women are underscreened for both cervical and breast cancer and immigrants also have a higher risk of underscreening for various cancers, regardless of time spent in Canada [7,26]. Tragically, these differences are further complicated by ongoing racial biases that lead to undertreatment of pain as well as reduced access to palliative care, a phenomenon best described as "not enough medical care for most of one's life, too much care at the end-of-life" [7,26].

Moreover, numerous studies highlight the service gap that exists for many Indigenous communities who often have to access distant locations for care which conflict with their values, beliefs, languages, and expectations [14,19]. Many Indigenous patients and families report that their spiritual and emotional needs are unmet in palliative care due to a lack of cultural safety, despite it being a priority [14,19].

Furthermore, it is important to recognize how people experience the intersecting impacts of racism with poverty, mental illness, substance use, and homelessness [24]. More than 235,000 Canadians experience homelessness annually (up to 35 k on any given night) and, as a result, are at increased risk of cancer-related deaths, shorter lifespan, and worse overall health outcomes [24,27–29]. Indigenous and other racially marginalized populations are overrepresented—only 0.5% of the population is Indigenous, but 15% are experiencing homelessness [30]. In Canada, 1 in 5 racially marginalized families will experience poverty compared to 1 in 20 non-racially marginalized families [31]. Furthermore, people experiencing homelessness have been noted to be 28 times more likely to have hepatitis C, and four times more likely to have cancer [32]. Meanwhile, life expectancies for unhoused populations have been estimated at between 34 and 47 years old, which is typically half that of those who have homes [33]. For these reasons, developing solutions to mitigate the obstacles these populations face in accessing palliative care requires an understanding of not only the intersectionality that shapes a patient's life, but a commitment to advancing social justice and human rights.

## 3. Examples of Current Initiatives Reducing Barriers in the Canadian Context

An example of an initiative that aims to reduce barriers is a website accessed through the Canadian Virtual Hospice called "Living My Culture" [34]. This website features people of various cultural backgrounds speaking in their native languages on their culture's beliefs, values, and practices around serious illness and end-of-life. For example, in a video under the section "Traditions, Rituals and Spirituality", a man from Hong Kong discusses how palliative care units and hospices are a new concept for many from his cultural background since the cultural norm he is accustomed to encompasses dying at home surrounded by family [34]. This demonstrates how a lack of knowledge of one's cultural context in navigating conversations such as location preference for end-of-life care can contribute to barriers in communication and understanding. This website aims to improve the quality of palliative care that these various groups receive by appropriately equipping providers with information and tools to improve cultural safety [34].

Another example is the presence of culturally appropriate long-term care homes. In Toronto, Yee Hong Long-Term Care (LTC) is one of the largest not-for-profit organizations in Canada that has served Chinese and other Asian seniors since 1994 to mitigate the language and cultural barriers these populations experienced in Toronto facilities [35]. Although other ethno-culturally focused LTC homes exist across the province, they only constitute 8% of the total [36]. These homes also tend to have the longest waitlists, which highlights the enormous gaps that continue to exist in the current system [36].

In addition, clinical models that aim to directly remove barriers for patients facing multiple structural vulnerabilities have been developed. One example is the PEACH (Palliative Education And Care for the Homeless) Program, an initiative developed through a partnership between Toronto-based Inner City Health Associates, Kensington Health and Toronto Central Home and Community Care Support Services. This mobile, street and

shelter-based palliative care program aims to provide low-barrier care for unhoused people through an anti-racist and anti-oppressive approach to care. With similar models now in operation across Canada in cities like Victoria, Edmonton and Calgary, programs like these make caring for racially marginalized people and uninsured populations a priority through their health justice focus [27,32,33].

## 4. Improving Equity at the Micro-, Meso-, and Macro-Levels

Although the aforementioned initiatives are an important component of improving health equity, the delivery of palliative care in Canada continues to be fragmented and siloed [37]. In the following, some recommendations for action at the micro-level (health worker and patient), meso-level (organizations and their communities), and macro-level (policy and politics) will be described for palliative care [38]. Most importantly, leadership and engagement by underrepresented cultural or racial groups and community stakeholders is required at all stages [19].

At the micro-level, increasing education and knowledge of cultural safety amongst providers is necessary to improve communication and informed decision-making around serious illness and end-of-life with patients and their loved ones. Providers can deepen their understanding of their own implicit bias by reflecting on how one's privilege, biases, emotions, and experiences shape one's values, especially with regard to serious illness or end-of-life discussions [4,15]. Providers can also deepen their knowledge of language barriers, cultural differences, and care options to improve overall communication and care decisions with patients of various backgrounds [8]. Examples of modalities that can be utilized to support this kind of knowledge acquisition include grand rounds, newsletters, ongoing curricula, modules, and quality improvement initiatives, which must be tied to meaningful action [7].

At the meso-level, health providers and health leadership must be more racially representative of the communities who require improved access and quality of palliative care services. Organizations can support and fund initiatives including providing anti-racism education for all providers and administrators, reviewing hiring practices to ensure diversity and inclusion, and ensuring the availability of interpretation services, health navigators and peer workers for marginalized populations [7]. Moreover, racially marginalized participants are underrepresented in most palliative care studies due to the lack of representation at higher levels of academia, particularly amongst the publishing industry and editorial boards [4]. Recruitment of racially diverse investigators can improve research which examines racial inequalities, and there must be support at the organizational level to address recognized disparities in the literature.

At the macro-level, system-level shifts in policy are required to ensure that cultural safety and anti-racism are at the core of healthcare policy and truly inclusive of populations historically excluded [5,24]. First, in order to measure the impacts of systemic racism on palliative care, collection of race-based data is required [37]. In the latest Canadian Institute for Health Information (CIHI) Palliative Care report from April 2023, although it was discussed that some people experience greater barriers in accessing palliative care, data did not include patient characteristics, such as race or language, which are key determinants in accessing such care [39]. While national and provincial frameworks reflect the aspirations of multiple levels of government to improve access to palliative care, ongoing disparities reflect a need for better integration and delivery of palliative care services for racially marginalized communities [40,41]. Health systems should recognize that improving equity in palliative care not only requires adequate funding for services, but a thorough understanding of how these services are designed and delivered and who may be excluded [24,42].

## 5. Conclusions

Current research in Canada has found that certain racially marginalized and immigrant populations are more likely to receive aggressive end-of-life care, have limited knowledge

of palliative care or advanced care planning, and experience challenges accessing culturally safe palliative care. These inequities stem from power imbalances that are inherent in our healthcare system and require identifying and changing the structures, policies and practices that perpetuate racism, discrimination, and lack of cultural safety. As healthcare providers, researchers and leaders, we all have a responsibility to advocate for the needs of diverse patients and their loved ones as they navigate serious illness and end-of-life.

**Author Contributions:** S.B., A.A. and N.D. contributed equally to the acquisition of articles and research publications for inclusion in this commentary. All authors contributed equally to the development of the commentary's first draft and editing of subsequent revisions. All authors have read and agreed to the published version of the manuscript.

**Funding:** This research received no external funding.

**Conflicts of Interest:** The authors declare no conflict of interest.

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
