# Peer review of "From Cultural Safety to Anti-Racism: Reflections on Addressing Inequities in Palliative Care"

_curroncol, doi:10.3390/curroncol30090575_

Round 1
Reviewer 1 Report
The article is properly reported.
The first time you include acronyms within the text, you have to write them in full. After that, you should report them as abbreviations only.
Just considering the type of article, it is a little long, so it should be more concise in all sections.
Conclusions is long, it should be summarized and please only present key findings
Impact of the COVID-19 Pandemic on Colorectal Cancer Screening: a Systematic Review and Psychological Distress among Cancer Patients during COVID-19 Pandemic in the World: A Systematic Review and Cultural safety strategies for rural Indigenous palliative care: a scoping review can be useful to improve the discussion.
Author Contributions and Funding is missed.
Please report at the end of the manuscript, before references, the following statement (in this order): Financial support: Ethics approval and consent to participate: Consent for publication: Availability of data and material: Conflict of interests: Funding: Authors Contributions:
Author Response
Point 1: The first time you include acronyms within the text, you have to write them in full. After that, you should report them as abbreviations only.
Response 1: We have scanned the paper for acronyms and ensured they are written in full at first appearance and abbreviated thereafter.
Point 2: Just considering the type of article, it is a little long, so it should be more concise in all sections. Conclusions is long, it should be summarized and please only present key findings
Response 2: Each section has been shortened and made more concise in an effort to reduce redundancy. The conclusion has been shortened.
Point 3: “Impact of the COVID-19 pandemic on Colorectal cancer screening: a systematic review” and “Psychological distress among cancer patients during COVID-19 pandemic in the world: a systematic review” and “cultural safety strategies for rural indigenous palliative care: a scoping review” can be useful to improve the discussion.
Response 3: Thank you for these article suggestions. Although we read all three, we feel that the last article suggestion is most in keeping with our discussion and has been referenced in the paper. The first two suggestions are out of the scope of this paper as they do not distinctly mention the health disparities which we present in this commentary.
Point 4: Author Contributions and Funding is missed. Please report at the end of the manuscript, before references, the following statement (in this order): Financial support: Ethics approval and consent to participate: Consent for publication: Availability of data and material: Conflict of interests: Funding: Authors Contributions
Response 4: This has been addressed.
Reviewer 2 Report
It is a very interesting commentary on unfair practices directed toward some disadvantaged groups, written, apparently, by professionals involved in palliative care in Canada. I appreciate very much that this problem has been raised, and I would recommend publishing the manuscript. However, I think, a few improvements could make it more sound for the general public.
First, could the authors use a first-hand opinion about the situation in palliative care in Canada? I mean opinions expressed by members of the groups mentioned in the commentary. Perhaps there are national survey data on how Indigenous people, other ethnic groups, immigrants, etc., perceive their situation in this respect and what they really need and want as regards palliative care. Or it might even be interviews with them in the media – it would help to understand the problem by readers.
Second, the authors, while describing current initiatives to reduce barriers, give some examples with links to websites, such as “Living My Culture”, “Yee Hong” and “PEACH”. I went through these websites and found them helpful in understanding cultural issues related to palliative care. I think that some excerpts from these websites could be used to illustrate problems from the recipients’ point of view.
Author Response
Point 1: First, could the authors use a first-hand opinion about the situation in palliative care in Canada? I mean opinions expressed by members of the groups mentioned in the commentary. Perhaps there are national survey data on how Indigenous people, other ethnic groups, immigrants, etc., perceive their situation in this respect and what they really need and want as regards palliative care. Or it might even be interviews with them in the media – it would help to understand the problem by readers.
Response 1: To address this suggestion, we further elaborated on a study done in Ontario called “Perceptions of palliative care in a South Asian community: findings from an observational study.” In this study, participants completed a survey which found that 70% of South Asians had not heard of palliative care, only 27% thought that it represented their cultural values, and a vast majority expressed a desire for more information on palliative care altogether. We also cited and discussed the paper “Cultural safety strategies for rural Indigenous palliative care: a scoping review” as recommended by Reviewer 1, and outlined that many Indigenous patients and families report unmet spiritual and emotional needs at end of life, despite it being a priority in their care. Finally, we referenced from My Virtual Hospice a patient perspective on how end-of-life discussions can have different meanings based on the cultural context.
Point 2: Second, the authors, while describing current initiatives to reduce barriers, give some examples with links to websites, such as “Living My Culture”, “Yee Hong” and “PEACH”. I went through these websites and found them helpful in understanding cultural issues related to palliative care. I think that some excerpts from these websites could be used to illustrate problems from the recipients’ point of view.
Response 2: An excerpt was taken from the Living My Culture page from Canadian Virtual Hospice to help provide context for different cultural barriers which may present in palliative care. Specifically, the example discusses a man who shares that back home in Hong Kong, palliative care units and hospices are a new concept for many people from China since the cultural norm he was accustomed to was to die at home surrounded by family. We use this example to illustrate how knowledge of differences in end-of-life care across various cultures can promote cultural safety, trust, and improved communication.
Reviewer 3 Report
This commentary explores reasons for low use of palliative care among persons of color, particularly those of African and South Asian descent, in Ontario and more generally.
The authors make the case that differences in accessibility of palliative care, lack of culturally responsive palliative care, later stages at diagnosis, and racially insensitive approaches among clinicians rather than culturally-driven differences in care needs or preferences explain these utilization patterns. They also not data gaps as an important factor in understanding and addressing these patterns.
The authors refer to racial and ethnic groups (indigenous persons and persons of African, Asian, and Pacific Island descent) as "racialized groups." This will be deeply offensive to readers in many countries who understand racial classification as a social/political process and view "whites" as also belonging to a "racialized" group.....In general, the authors seems to struggle with naming groups and should say something explicit about this.
The paper lacks a conceptual framework for talking about systemic racism and its multi-level expressions. They offer a 3 level (micro, meso, and macro levels) framework for recommendations but they did not fully develop this framework in their review of evidence.
The evidence review seems cursory and there is no explication of criteria for inclusion or exclusion of evidence. If this is not a systematic review of the literature, the authors need to offer some other set of criteria for selecting evidence. The authors draw materials from UK, US, and Canadian research but never address differences in DEI requirements, strategies, and outcomes across the countries. The comparisons may well be instructive and they need to be contextualized.
Oddly, at the micro/personal level they recommend that "....increasing.....education among providers must be streamlined." I don't think there was any evidence offered that this equitable care training of providers has been offered systematically, is broadly implemented, and has know impacts......I don't understand what "streamlined" means in this context or if it is the same thing as making institutional accreditation and professional licensing contingent on evidence of meeting a standard of knowledge and practice related to racial/ethnic differences.
The authors also offer recommendations regarding health care institutions and broader policy making. The review does not indicate to what extent current law or professional practice standards require attention to diversity and inclusion or equitable treatment in Canada; how well such laws and standards are being implemented or the political process required to update these laws to address diversity among patients.
See my notes about the use of "racialized".....this construct is used quite differently in US English than by the authors and needs to be explained for a worldwide audience.
Similarly, the authors describe utilization of end-of-life intensive medical interventions as "use" by members of specific groups......I think this misses the idea that in the end-of-life context, most demand for care is professionally induced----it is what people are given, not what they have elected to use.
Author Response
Point 1: The authors refer to racial and ethnic groups (indigenous persons and persons of African, Asian, and Pacific Island descent) as "racialized groups." This will be deeply offensive to readers in many countries who understand racial classification as a social/political process and view "whites" as also belonging to a "racialized" group.....In general, the authors seems to struggle with naming groups and should say something explicit about this.
Response 1: Although in the Canadian-specific context, the term “racialized” as defined by the Ontario Human Rights Commission means “communities that face racism,” reviewer 3 provides helpful feedback that there is no worldwide consensus on the definition of this term, and as such, can cause confusion. As a result, all instances in the commentary that use the term “racialized” have been altered to distinctly list the communities or groups being referenced to, or we use the term “racially marginalized” in attempt to reference non-white communities.
Point 2: The paper lacks a conceptual framework for talking about systemic racism and its multi-level expressions. They offer a 3 level (micro, meso, and macro levels) framework for recommendations but they did not fully develop this framework in their review of evidence.
Response 2: We kindly ask for clarification on this point. Given our expertise and extensive work in this area, it is unclear how to incorporate this framework in our review of evidence. The use of micro, meso, and macro levels as a framework for discussing recommendations for action are referenced from the well-renowned article “Practicing Social Accountability: From Theory to Action.” This article discusses how the individual patient level (micro), community level (meso), and macro level (politics and public policy) must be addressed in order to ensure care is “equitably accessible to everyone and responsive to patient, community, and population health needs,” and is frequently used in conversations around social accountability of health equity.
Point 3: The evidence review seems cursory and there is no explication of criteria for inclusion or exclusion of evidence. If this is not a systematic review of the literature, the authors need to offer some other set of criteria for selecting evidence. The authors draw materials from UK, US, and Canadian research but never address differences in DEI requirements, strategies, and outcomes across the countries. The comparisons may well be instructive and they need to be contextualized.
Response 3: This paper is written as a commentary and not a systematic review of literature. As such, evidence used in this commentary did not undergo rigorous criteria for selection but rather relevant literature was referenced to help support the views discussed in the commentary. This commentary does not aim to discuss differences in DEI strategies and thus evidence which was included in this article outside of the Canadian context was redacted. A vast majority of the references used in this commentary are from Canada and on this revision we have limited research and data outside of Canada as comparing DEI strategies is not in scope with our article.
Point 4: Oddly, at the micro/personal level they recommend that "....increasing.....education among providers must be streamlined." I don't think there was any evidence offered that this equitable care training of providers has been offered systematically, is broadly implemented, and has know impacts......I don't understand what "streamlined" means in this context or if it is the same thing as making institutional accreditation and professional licensing contingent on evidence of meeting a standard of knowledge and practice related to racial/ethnic differences.
Response 4: Given the unclear use of the term “streamlined” in this context, we have changed the wording in this to “is necessary.”
Point 5: The authors also offer recommendations regarding health care institutions and broader policy making. The review does not indicate to what extent current law or professional practice standards require attention to diversity and inclusion or equitable treatment in Canada; how well such laws and standards are being implemented or the political process required to update these laws to address diversity among patients.
Response 5: In this revision, we have significantly revised the recommendations section. Specifically, we list that although national and provincial palliative care frameworks exist (which discuss mandates on equity, diversity, and inclusion), the existance of ongoing disparities highlights the gaps that persist. We discuss the lack of collection of race-based data as an ongoing barrier, and reorient the recommendations on the need to design and deliver services that are mindful of who has been historically excluded.
Point 6: See my notes about the use of "racialized".....this construct is used quite differently in US English than by the authors and needs to be explained for a worldwide audience.
Response 6: All instances of the term “racialized” have been reworded in the commentary to improve clarity.
Point 7: Similarly, the authors describe utilization of end-of-life intensive medical interventions as "use" by members of specific groups......I think this misses the idea that in the end-of-life context, most demand for care is professionally induced----it is what people are given, not what they have elected to use.
Response 7: The term “use” has been changed to “receive” to improve clarity.
Reviewer 4 Report
The authors of this manuscript attempt to show that minorities may not have access to health resources. Taking the COVID-19 pandemic as an example of impaired health measures in special populations in Canada, they take it one step further describing potential inequities in palliative care.
A few crucial points are:
- Although claiming in favor of the minorities, the authors tend to be denunciatory to national policies.
- The claim in the title about "how to make palliative care more equitable" are not clearly addressed. There are 3 levels described, but this part seems quite fuzzy and clearly needs improvement.
- There are a lot of References to webpages, and the References to Journals must support your text.
- The article seems to be outside the scope of the Journal that is submitted to.
Author Response
Point 1: Although claiming in favor of the minorities, the authors tend to be denunciatory to national policies.
Response 1: Data supports that despite national policies to improve health equity, inequities and disparities in health outcomes continue to impact populations that are racially marginalized or experience discrimination. We kindly ask for further clarification into the section of this commentary that is denunciatory toward national policies, as our aim is to recognize that despite attempts to improve equity, gaps and disparities in care are ongoing and that improvement is not only possible but necessary, as highlighted in the numerous cited examples of evidence.
Point 2: The claim in the title about "how to make palliative care more equitable" are not clearly addressed.
Response 2: Thank you for this feedback. In attempt for our title to be more reflective of the content within the commentary, the title has been changed to “From cultural safety to anti-racism: Reflections on addressing inequities in palliative care.”
Point 3: There are 3 levels described, but this part seems quite fuzzy and clearly needs improvement.
Response 3: This section has been completely revised, shortened, condensed, and edited for clarity and directness.
Point 4: There are a lot of References to webpages, and the References to Journals must support your text.
Response 4: A mix of journal references and webpages were used to help support the discussion in this commentary and all references have been checked to ensure they adequately support the text used in the commentary.
Point 5: - The article seems to be outside the scope of the Journal that is submitted to.
Response 5: From our understanding, we were invited to write a commentary on the topic of cultural safety and equity in palliative care for Current Oncology. Revisions have been made by all reviewers in an attempt to improve clarity in our discussion on this topic.
Round 2
Reviewer 1 Report
-
Author Response
Thank you
Reviewer 3 Report
I think the authors have done an excellent job in editing this commentary and have addressed all my concerns that made any sense. I really appreciate the update of the recommendations section. I think the paper is well-written and well-edited at this point.
Author Response
Thank yiu
Reviewer 4 Report
The authors have already made suggested alterations. The overall sense of the manuscript seems to be more balanced and well described and cited.
Please consider adding essential conflict of interest statement regarding the organization that is mentioned in the text and an author is Head of.
Author Response
Thank you. Although one author has an affiliation that is mentioned we believe there are no conflicts of interest.